# Transition Rates for $3s3p^2\ ^4P - 3s3p4s\ ^4P^o$ Transitions in Al I

**Charlotte Froese Fischer** [1,*] and **James F. Babb** [2,*]

[1] Department of Computer Science, University of British Colombia, 2366 Main Mall, Vancouver, BC V6T1Z4, Canada

[2] Institute for Theoretical Atomic, Molecular, and Optical Physics (ITAMP), Center for Astrophysics | Harvard & Smithsonian, 60 Garden St., MS 14, Cambridge, MA 02138, USA

[*] Correspondence: cff@cs.ubc.ca (C.F.F.); jbabb@cfa.harvard.edu (J.F.B.); Tel.: +1-604-225-5147 (C.F.F.); +1-617-496-7612 (J.F.B.)

**Abstract:** Fully relativistic calculations have been performed for two multiplets, $3s3p^2\ ^4P$ and $3s3p4s\ ^4P^o$, in Al I. Wave functions were obtained for all levels of these multiplets using the GRASP programs. Reported are the E1 transitions rates for all transitions between levels of these multiplets. Transition energies and transition rates are compared with observed values and other theory. Our calculated transition rates are smaller by about 10% than observed rates, reducing a large discrepancy between earlier calculations and experiments.

**Keywords:** atomic spectra; energy levels; transition probabilities; wavelengths; aluminum

---

## 1. Introduction

Atomic spectra are of vital importance as plasma diagnostics and reliable wavelengths and transition probabilities are essential for applications. Recently, Hermann et al. [1] deduced transition probabilities between the fine structure lines connecting the $3s3p^2\ ^4P$ and $3s3p^2\ ^4P^o$ multiplets from emission coefficients measured in laser ablation of aluminum in argon. The transition probabilities obtained were roughly a factor of two larger than those listed in the semi-empirical calculations of Kurucz and Peytremann [2], and no other values were found in existing tabulations. The fine structure lines connecting the $3s3p^2\ ^4P$ and $3s3p^2\ ^4P^o$ multiplets occur in the same general wavelength region (305–310 nm) and with comparable strength as the well-studied (see [3] and the references therein) fine structure lines connecting the $3s^23p\ ^2P$ and $3s^23d\ ^2D$ multiplets, making the discrepancy of concern for applications in the UV wavelengths.

Multiconfiguration Dirac–Hartree–Fock (MCDHF) and relativistic configuration interaction (RCI) calculations have been performed by Papoulia et al. [3] for 28 states in neutral Al. The configurations of interest were $3s^2nl$ for $n = 3, 4, 5$ with $l = 0–4$, as well as $3s3p^2$ and $3s^26l$ for $l = 0, 1, 2$. Lifetimes and transition data for radiative electric dipole (E1) transitions were reported. There was a significant improvement in accuracy, in particular for the more complex system of neutral Al I, which may prove useful for astrophysical applications to Al abundance determinations in stars. Omitted were the levels of the $3s3p4s\ ^4P^o$ multiplet, which lies above the first $3s^2$ ionization limit [4].

This paper reports transition rates for all E1 transitions between the $3s3p^2\ ^4P$ and $3s3p4s\ ^4P^o$ multiplets using the variational multiconfiguration Dirac–Hartree–Fock (MCDHF) method [5], as implemented in the GRASP programs [6]. The accuracy of the results is based on the accuracy of the theoretically-predicted transition energies compared with available measurements, as well as the agreement between length and velocity rates.

## 2. Underlying Theory

In the MCDHF method [5], the wave function $\Psi(\gamma PJM_J)$ for a state labeled $\gamma PJM_J$, where $J$ and $M_J$ are the angular quantum numbers and $P$ the parity, is expanded in antisymmetrized and coupled configuration state functions (CSFs):

$$\Psi(\gamma PJM_J) = \sum_{j=1}^{M} c_j \Phi(\gamma_j PJM_J). \tag{1}$$

The labels $\{\gamma_j\}$ denote other appropriate information about the CSFs, such as orbital occupancy and the coupling scheme. The CSFs are built from products of one-electron orbitals, having the general form:

$$\psi_{n\kappa,m}(\mathbf{r}) = \frac{1}{r} \begin{pmatrix} P_{n\kappa}(r)\chi_{\kappa,m}(\theta,\varphi) \\ \iota Q_{n\kappa}(r)\chi_{-\kappa,m}(\theta,\varphi) \end{pmatrix}, \tag{2}$$

where $\chi_{\pm\kappa,m}(\theta,\varphi)$ are two-component spin-angular functions. The expansion coefficients and the radial functions are determined iteratively. In the present work, the Dirac–Coulomb Hamiltonian $\mathcal{H}_{DC}$ was used [5], which included a correction for the finite size of the nucleus.

The radial functions $\{P_{n\kappa}(r), Q_{n\kappa}(r)\}$ were determined numerically as solutions of differential equations,

$$w_a \begin{bmatrix} V(a;r) & -c\left[\frac{d}{dr}-\frac{\kappa_a}{r}\right] \\ c\left[\frac{d}{dr}+\frac{\kappa_a}{r}\right] & V(a;r)-2c^2 \end{bmatrix} \begin{bmatrix} P_a(r) \\ Q_a(r) \end{bmatrix} = \sum_b \epsilon_{ab}\, \delta_{\kappa_a\kappa_b} \begin{bmatrix} P_b(r) \\ Q_b(r) \end{bmatrix}, \tag{3}$$

where $V(a;r) = V_{nuc}(r) + Y(a;r) + \bar{X}(a;r)$ is a potential consisting of nuclear, direct, and exchange contributions arising from both diagonal and off-diagonal $\langle\Phi_\alpha|\mathcal{H}_{DC}|\Phi_\beta\rangle$ matrix elements [5].

For a given set of radial functions, expansion coefficients $\mathbf{c} = (c_1,\dots,c_M)^t$ were obtained as solutions to the configuration interaction (CI) problem,

$$\mathbf{H}\mathbf{c} = E\mathbf{c}, \tag{4}$$

where $\mathbf{H}$ is the CI matrix of dimension $M \times M$ with elements

$$H_{ij} = \langle\Phi(\gamma_i PJM_J)|H|\Phi(\gamma_j PJM_J)\rangle. \tag{5}$$

Once self-consistent solutions have been obtained—sometimes referred to as the relativistic MCDHF or RMCDHF phase—an RCI calculation was performed using an extended Hamiltonian that included the transverse photon (Breit) and QED corrections. Wave functions from the latter Hamiltonian were used to compute the E1 transitions rates.

## 3. Systematic Procedures

Systematic procedures were used in which the orbital set used for defining the wave function expansion increased systematically within a correlation model.

The states of Al consist of a neon-like ($1s^2 2s^2 2p^6$) core and three valence electrons. Wave function expansions were obtained from single- and double- (SD) excitations from a multireference (MR) set that interacted significantly with the CSFs of interest. For the even multiplet, the MR set included the CSFs from $3s3p^2$, $3s3d^2$, and $3p^2 3d$ configurations and for the odd multiplet, CSFs from $3s3p4s$, $3p3d4s$, $3p^2 4p$, and $3s3d4p$. Because the odd multiplet was above the first ionization limit, all $3s^2 nl$ CSFs were removed. In this paper, the orbital sets that define the set of excitations were classified according to the

largest $nl$ of the orbital set when the latter were ordered globally by $n$ (the principal quantum number) and within $n$ by $l$ (the orbital quantum number). Thus, an $n = 3$ orbital set includes all orbitals up to $3s, 3p, 3d$ and all $n = 5f$ orbitals up to $5s, 5p, 5d, 5f$ ($5g$ not included).

Our first model was the valence correlation (VV) model, in which all excitations involved only valence electrons. $n = 3$ calculations were performed for an average energy functional of the lowest even parity $J = 1/2, 3/2, 5/2$ states. This calculation defined the core orbitals for all subsequent calculations. The $n = 4, 5f, 6f$ calculations each varied only the new orbitals. For the odd multiplet, the first calculation had orbitals up to $4s, 4p$, and $3d$, but is still referred to as an $n = 3$ calculation in this paper, with remaining sets being the regular $n = 4, 5f, 6f$ orbital sets. Table 1 shows the convergence of the fine-structure of the two multiplets and their separation.

**Table 1.** Convergence of the energy level structure for a valence correlation calculation is compared with observed data [4,7]. All results are in cm$^{-1}$.

| | $n = 3$ | $n = 4$ | $n = 5f$ | $n = 6f$ | Observed |
|---|---|---|---|---|---|
| $3s3p^2\ ^4P$ Fine structure | | | | | |
| $3s3p^2\ ^4P_{1/2}$ | 0 | 0 | 0 | 0 | 0.00 |
| $3s3p^2\ ^4P_{3/2}$ | 46 | 46 | 45 | 45 | 46.55 |
| $3s3p^2\ ^4P_{5/2}$ | 122 | 121 | 120 | 120 | 122.37 |
| $3s3p4s\ ^4P^o$ Fine structure | | | | | |
| $3s3p4s\ ^4P^o_{1/2}$ | 0 | 0 | 0 | 0 | 0.00 |
| $3s3p4s\ ^4P^o_{3/2}$ | 57 | 58 | 56 | 55 | 56.10 |
| $3s3p4s\ ^4P^o_{5/2}$ | 156 | 157 | 153 | 150 | 152.08 |
| Multiplet separation | | | | | |
| $3s3p4s\ ^4P^o_{1/2} - 3s3p^2\ ^4P_{1/2}$ | 33,213 | 33,029 | 32,724 | 32,696 | 32,671.05 |

The results of the converged valence correlation calculations, when compared with observation, suggest that the energy structure is not significantly affected by the core-valence (CV) that accounts for the polarization for the core. To confirm this conclusion, calculations were performed in which SD excitations included a single excitation from the $2p$-shell along with a single excitation of a valence electron. Wave function expansions were considerably larger and convergence a bit slower. Table 2 shows the convergence of the energy structure. The fine structure of the odd multiplet increased slightly and was in somewhat better agreement with the data from observation. At the same time, the transition energy for $^4P_{1/2} - ^4P^o_{1/2}$ for an $n = 7f$ calculation was not in as good agreement with observed as the $n = 6f$ valence correlation calculation reported in Table 1.

**Table 2.** Convergence of the energy level structure from a core-valence plus valence correlation calculation is compared with observed data [4,7]. All results are in cm$^{-1}$.

| | $n = 5f$ | $n = 6f$ | $n = 7f$ | Observed |
|---|---|---|---|---|
| $3s3p^2\ ^P$ Fine structure | | | | |
| $3s3p^2\ ^4P_{1/2}$ | 0 | 0 | 0 | 0.00 |
| $3s3p^2\ ^4P_{3/2}$ | 48 | 45 | 45 | 46.55 |
| $3s3p^2\ ^4P_{5/2}$ | 128 | 120 | 120 | 122.37 |
| $3s3p4s\ ^4P^o$ Fine structure | | | | |
| $3s3p4s\ ^4P^o_{1/2}$ | 0 | 0 | 0 | 0.00 |
| $3s3p4s\ ^4P^o_{3/2}$ | 60 | 56 | 56 | 56.10 |
| $3s3p4s\ ^4P^o_{5/2}$ | 167 | 155 | 153 | 152.08 |
| Multiplet separation | | | | |
| $3s3p^2\ ^4P_{1/2} - 3s3p4s\ ^4P^o_{1/2}$ | 33,142.58 | 32,934.21 | 32,838.33 | 32,671.05 |

## 4. Results

The wave functions from RCI expansions, determined using the Dirac–Coulomb–Breit-QED Hamiltonian, were used to compute the E1 transition rates for all transitions between these two multiplets. Table 3 reports the transition energy $\Delta E$ (cm$^{-1}$), the wavelength $\lambda$ (nm) in a vacuum, $A$ (µs$^{-1}$), and $gf$, in the length form for calculations of Table 1. Furthermore included is an indicator of accuracy $dT = (A_l - A_v)/\max(A_l, A_v)$, where $A_l$ and $A_v$ are transition rates from length and velocity forms, respectively. The average discrepancy between the two forms was 1.5%, and in all cases, the velocity form had a larger value than the length form.

**Table 3.** Ab initio electric dipole (E1) transition data for the $3s3p^2\ {}^4P$ to $3s3p4s\ {}^4P^o$ transition computed in the length form from valence correlation results.

| Upper | Lower | $\Delta E$ (cm$^{-1}$) | $\lambda$ (nm) | $A$ (µs$^{-1}$) | $gf$ | $dT$ |
|---|---|---|---|---|---|---|
| $3s3p4s\ {}^4P^o_{1/2}$ | $3s3p^2\ {}^4P_{1/2}$ | 32,696 | 305.84220 | 29.93 | 0.0840 | 0.016 |
| $3s3p4s\ {}^4P^o_{1/2}$ | $3s3p^2\ {}^4P_{3/2}$ | 32,650 | 306.27027 | 149.01 | 0.4191 | 0.017 |
| $3s3p4s\ {}^4P^o_{3/2}$ | $3s3p^2\ {}^4P_{1/2}$ | 32,752 | 305.32421 | 75.11 | 0.4199 | 0.014 |
| $3s3p4s\ {}^4P^o_{3/2}$ | $3s3p^2\ {}^4P_{3/2}$ | 32,706 | 305.75074 | 23.89 | 0.1339 | 0.015 |
| $3s3p4s\ {}^4P^o_{3/2}$ | $3s3p^2\ {}^4P_{5/2}$ | 32,631 | 306.44982 | 80.43 | 0.4529 | 0.017 |
| $3s3p4s\ {}^4P^o_{5/2}$ | $3s3p^2\ {}^4P_{3/2}$ | 32,801 | 304.86634 | 54.37 | 0.4546 | 0.014 |
| $3s3p4s\ {}^4P^o_{5/2}$ | $3s3p^2\ {}^4P_{5/2}$ | 32,726 | 305.56128 | 125.94 | 1.0577 | 0.015 |

The effect of core-valence is shown in Table 4 where results are reported both for the $n = 5f$ and $n = 7f$ calculation. The former was included because of the remarkable agreement in length and velocity forms, yet the transition energy (as shown in Table 2) was not in as good agreement with the observed as before. Since the transition rate is proportional to $(\Delta E)^3$, correcting the transition rate for this factor would introduce a 4.0% reduction. Indeed, the $n = 7f$ transition rates were smaller with the transition energy more accurate, but length and velocity were not in as good agreement.

**Table 4.** Ab initio electric dipole (E1) transition data for the $3s3p^2\ {}^4P$ to $3s3p4s\ {}^4P^o$ transition computed in the length form from valence and core-valence results.

| Upper | Lower | $\Delta E$ (cm$^{-1}$) | $\lambda$ (nm) | $A$ (µs$^{-1}$) | $gf$ | $dT$ |
|---|---|---|---|---|---|---|
| $n = 5f$ | | | | | | |
| $3s3p4s\ {}^4P^o_{1/2}$ | $3s3p^2\ {}^4P_{1/2}$ | 33,142 | 301.727 | 29.76 | 0.0812 | 0.004 |
| $3s3p4s\ {}^4P^o_{1/2}$ | $3s3p^2\ {}^4P_{3/2}$ | 33,093 | 302.170 | 148.19 | 0.4057 | 0.006 |
| $3s3p4s\ {}^4P^o_{3/2}$ | $3s3p^2\ {}^4P_{1/2}$ | 33,203 | 301.173 | 74.72 | 0.4064 | 0.001 |
| $3s3p4s\ {}^4P^o_{3/2}$ | $3s3p^2\ {}^4P_{3/2}$ | 33,154 | 301.615 | 23.88 | 0.1303 | 0.003 |
| $3s3p4s\ {}^4P^o_{3/2}$ | $3s3p^2\ {}^4P_{5/2}$ | 33,075 | 302.341 | 79.91 | 0.4380 | 0.008 |
| $3s3p4s\ {}^4P^o_{5/2}$ | $3s3p^2\ {}^4P_{3/2}$ | 33,261 | 300.652 | 53.83 | 0.4377 | 0.001 |
| $3s3p4s\ {}^4P^o_{5/2}$ | $3s3p^2\ {}^4P_{5/2}$ | 33,181 | 301.373 | 125.12 | 1.0223 | 0.003 |
| $n = 7f$ | | | | | | |
| $3s3p4s\ {}^4P^o_{1/2}$ | $3s3p^2\ {}^4P_{1/2}$ | 32,838 | 304.522 | 28.51 | 0.0793 | 0.051 |
| $3s3p4s\ {}^4P^o_{1/2}$ | $3s3p^2\ {}^4P_{3/2}$ | 32,792 | 304.947 | 141.94 | 0.3958 | 0.053 |
| $3s3p4s\ {}^4P^o_{3/2}$ | $3s3p^2\ {}^4P_{1/2}$ | 32,894 | 304.003 | 71.57 | 0.3966 | 0.049 |
| $3s3p4s\ {}^4P^o_{3/2}$ | $3s3p^2\ {}^4P_{3/2}$ | 32,848 | 304.426 | 22.85 | 0.1270 | 0.050 |
| $3s3p4s\ {}^4P^o_{3/2}$ | $3s3p^2\ {}^4P_{5/2}$ | 32,773 | 305.120 | 76.55 | 0.4274 | 0.055 |
| $3s3p4s\ {}^4P^o_{5/2}$ | $3s3p^2\ {}^4P_{3/2}$ | 32,946 | 303.523 | 51.61 | 0.4277 | 0.047 |
| $3s3p4s\ {}^4P^o_{5/2}$ | $3s3p^2\ {}^4P_{5/2}$ | 32,871 | 304.213 | 119.86 | 0.9979 | 0.050 |

In summary, valence correlation predicted the best transition energy and the best agreement in length and velocity for accurate transition energy.

## 5. Summary and Conclusions

Table 5 compares the predicted transition rates (based on observed transition energies or wavelengths in a vacuum, rather than computed transition energies as in Table 4), with values derived from observations by Hermann et al. [1] and the values reported by Kurucz and Peytremann [2]. The latter used a semi-empirical approach in which Slater parameters were determined empirically from observed energy levels and transition probabilities calculated by the use of scaled Thomas–Fermi–Dirac wave functions. As seen in Table 5, the present predicted transition rates are about 10% smaller than observed values, whereas the Kurucz and Peytremann values are about a half those of the observed rates. Thus, the discrepancy between theory and experiment has been reduced significantly.

**Table 5.** Comparison of the transition rates computed from valence correlation calculations (present) and observed wavelengths (in a vacuum) from NIST [4,7] with observed rates from Hermann et al. [1] and values reported by Kurucz and Peytremann [2].

| Upper | Lower | $\lambda$ (nm) | | $A$ ($\mu$s$^{-1}$) | | |
|---|---|---|---|---|---|---|
| | | NIST | Present | Present | Hermann | Kurucz |
| $3s3p4s\ ^4P^o_{1/2}$ | $3s3p^2\ ^4P_{1/2}$ | 305.9924 | 305.8422 | 29.89 | 34. | 18.3 |
| $3s3p4s\ ^4P^o_{1/2}$ | $3s3p^2\ ^4P_{3/2}$ | 306.4290 | 306.2703 | 148.80 | 160. | 89.2 |
| $3s3p4s\ ^4P^o_{3/2}$ | $3s3p^2\ ^4P_{1/2}$ | 305.4679 | 305.3242 | 74.99 | 78. | 44.9 |
| $3s3p4s\ ^4P^o_{3/2}$ | $3s3p^2\ ^4P_{3/2}$ | 305.9029 | 305.7507 | 23.85 | 28. | 14.2 |
| $3s3p4s\ ^4P^o_{3/2}$ | $3s3p^2\ ^4P_{5/2}$ | 306.6144 | 306.4498 | 80.32 | 90. | 47.7 |
| $3s3p4s\ ^4P^o_{5/2}$ | $3s3p^2\ ^4P_{3/2}$ | 305.0073 | 304.8663 | 54.28 | 59. | 32.1 |
| $3s3p4s\ ^4P^o_{5/2}$ | $3s3p^2\ ^4P_{5/2}$ | 305.7144 | 305.5613 | 125.75 | 140. | 75.0 |

**Author Contributions:** The authors C.F.F. and J.F.B. contributed jointly to conceptualization, methodology, validation, formal analysis, writing–original draft preparation, and writing–review and editing.

**Funding:** This research was funded by Canada NSERC Discovery Grant 2017-03851 (CFF) and US NSF Grant No. PHY-1607396 (JFB). The APC was funded by MDPI.

**Acknowledgments:** The authors (CFF and JB) acknowledge support, respectively, from the Canada NSERC Discovery Grant 2017-03851 and from ITAMP, which is supported in part by Grant No. PHY-1607396 from the NSF to Harvard University and the Smithsonian Astrophysical Observatory.

**Conflicts of Interest:** The authors declare no conflict of interest. The funders had no role in the design of the study; in the collection, analyses, or interpretation of data; in the writing of the manuscript, or in the decision to publish the results.

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
