# Peer review of "Transition Rates for 3s3p2 4P–3s3p4s 4Po Transitions in Al i"

_atoms, doi:10.3390/atoms7020054_

Round 1
Reviewer 1 Report
Refree's report on "Transition rates for 3s3p2 4P - 3s3p4s 4Po transitions in Al I" by C. F. Fischer and J.F. Babb.
This paper reports the E1 transition rates from the 3s3p4s 4Po upper states to the 3s3p2 4P lower states calculated by the variational Multiconfiguration Dirac-Hatree-Fock method using the GRASP2018.
The underlying theory and the calculation detailes for configurations sets, valence-valence (VV) correlation vs. core-valence (CV) correlation, their convergence for energy level structure and the numerical accuracy of transition rate calcuation by comparing the length and the velocity gauge calcuation are well described.
Their VV correlation results showing better accuracies in energy levels and numerical agreement between the length and the velocity gauges than the CV correlation results are ~10% lower than the observed transtion rates by Herman et al. while the semi-empirical predictions by Kurucz et al. are ~50% lower than the experimental results. This is a significantly improved agreement between the theory and the experiment .
Here are just minor comments.
In the Table 5, the listed "present" A values are different from those in the Table 3.
Are those in the Table 5 for the velocity guage values unlike in the Table 3 or any other reasons?
There is a typo such as "Table 1" in the Results section of page 3 which should be "Table 3".
The captions for the Table 3 and 4 are same but different VV and CV correlations need to be remarked in the table captions.
Author Response
Reply to Reviewer 1. (Fischer & Babb).
Thank you for the valuable comments, which we've implemented in
the revised manuscript.
1. The results in Table 5 were calculated with observed transition energies
and thus slightly differ from Table 3. We've added a comment to this
effect to the first sentence of Sec. 5 (Summary and Conclusions).
2. Typo fixed.
3. We've added descriptions to the captions of Tables 3 and 4 as suggested.
Reviewer 2 Report
Recommendations for the authors.
1. Consider adding a line to the abstract with a summary of the key results.
2. Consider moving the second paragraph of the introduction to the beginning of the introduction.
3. Line 23: Consider re-writing as: “… The accuracy of the calculated results is established by comparing the transition energy with available measurements.”
4. Line 48: Consider re-writing as: “… (5g is not included)”
5. Line 55: Consider removing “All results are reported in cm-1.”
6. Caption of Table 1: re-write as: “ … is compared with observed data [2,7]”
7. Consider adding a separated paragraph for the discussion of Table 5.
8. Consider adding a Summary or Conclusion section for the summary or conclusion.
Author Response
Reply to Reviewer 2 (Fishcher & Babb)
Thank you for the valuable comments, which we've implemented in
the revised manuscript.
We have modified the abstract and added a short summary sentence.
Done.
Done. Note that this is now line 28 in the revised manuscript.
Done.
Done.
Done.
Done. See new line 87.
Done. See new line 86.